# Human Papillomavirus Vaccines: An Updated Review

**DOI:** 10.3390/vaccines8030391

**Published:** 2020-07-16

**Authors:** Liqin Cheng, Yan Wang, Juan Du

**Affiliations:** Department of Microbiology, Tumor and Cell Biology (MTC), Karolinska Institutet, 17177 Stockholm, Sweden; liqin.cheng@ki.se (L.C.); yan.wang@ki.se (Y.W.)

**Keywords:** human papillomavirus, vaccine, Cervarix, Gardasil, Gardasil 9

## Abstract

Human papillomavirus (HPV) vaccines, which were introduced in many countries in the past decade, have shown promising results in decreasing HPV infection and related diseases, such as warts and precancerous lesions. In this review, we present the updated information about current HPV vaccines, focusing on vaccine coverage and efficacy. In addition, pan-gender vaccination and current clinical trials are also discussed. Currently, more efforts should be put into increasing the vaccine’s coverage, especially in low- and middle-income countries. Provision of education on HPV and vaccination is one of the most important methods to achieve this. Vaccines that target HPV types not included in current vaccines are the next stage in vaccine development. In the future, all HPV-related cancers, such as head and neck cancer, and anal cancer, should be tracked and evaluated, especially in countries that have introduced pan-gender vaccination programs. Therapeutic vaccines, in combination with other cancer treatments, should continue to be investigated.

## 1. Introduction

Human papillomavirus (HPV) infection, one of the most common sexually transmitted diseases, is associated with cancers such as cervical cancer, head and neck squamous cell carcinoma (HNSCC), and anal cancer [1,2,3]. To date, more than 200 HPV types have been identified [4,5]. HPV infections are transmitted primarily through skin-to-skin or skin-to-mucosa contact. Some HPV types mainly infect cutaneous tissues and induce warts, while other HPV types mainly target mucosal tissues of cervical and oral tracts [1,6]. Depending on the oncogenic potential, various mucosal HPV types are categorized as either high-risk HPV/oncogenic HPV types, which can be potentially carcinogenic, e.g., HPV16, 18, 31, and 33, or low-risk HPV/non-oncogenic HPV types, which are mostly found in warts, e.g., HPV6 and 11 [7,8]. Fortunately, three HPV vaccines were introduced against up to nine HPV types, showing strong protection against cervical infections caused by these HPV types as well as condylomas and some HPV-related cancers [9,10,11,12,13,14,15]. There are great reviews on HPV infections and their relation to different cancers, including oropharyngeal, vaginal, vulvar, penile, and anal cancers [1,16,17,18]. In this review, we focus on HPV vaccines and their effects. 

## 2. HPV Vaccine Coverage

Quadrivalent HPV vaccine, Gardasil (Merck & Co., Kenilworth, NJ, USA), is the first commercially available HPV vaccine licensed by the United States Food and Drug Administration (FDA), in 2006. The bivalent HPV vaccine, Cervarix (GSK, Brentford, UK) was approved by the European Medicines Agency (EMA) in 2007 and by the FDA in 2009 [19]. Cervarix protects against the most common oncogenic genotypes of HPV (types 16 and 18), which cause around 70% of cervical cancers [20]. Gardasil, in addition to HPV16 and 18, also targets HPV6 and 11, which cause around 90% of genital warts [21]. In 2014, a nine-valent vaccine, Gardasil 9 (Merck & Co., Kenilworth, NJ, USA), was licensed by the FDA, which offers protection against HPV6, 11, 16, 18, 31, 33, 45, 53, and 58. The five additional types covered by Gardasil 9 could cover HPV types related to another 20% of cervical cancer cases; thus, Gardasil 9 has the potential to protect against approximately 90% of cervical cancers [22]. A comparative modeling analysis predicted that if the global strategy of combined intensive scaled-up HPV vaccination and twice-lifetime screening is achieved, the incidence of 97% cervical cancers would be reduced by 2100 [23].

In general, HPV national programs cover about 30% of the global target population, with low full-dose coverage in many regions [24]. The HPV vaccine coverage is significantly higher in high-income countries, where about 32% of females aged 10–20 years received the full-dose vaccination by 2014 [24,25]. HPV coverage is more than 60% in countries such as Australia, Denmark, and Sweden [24,26,27]. Most low- and middle-income countries remain unprotected, only about 1% of adolescent females in low-income countries received a full course of HPV vaccines [24]. Fortunately, HPV vaccine was successfully introduced in some low- and middle-income countries’ national routine immunization schedules and achieved high coverage, such as in Bhutan and Rwanda [26,28]. The HPV-vaccination gap also exists between urban and rural residents within low- and middle-income countries. For example, although 65.3% of children are fully immunized in India, the HPV coverage is only 2% in rural villages of Uttar Pradesh, with 72% vaccine coverage concentrated in urban areas [29]. As more than 80% of cervical cancer deaths occur in low- and middle-income countries, the implementation of the HPV vaccine is urgently needed for public health intervention in these countries [25]. 

## 3. Mechanisms of Vaccinations

Currently, the licensed HPV vaccines are developed based on a virus-like particle (VLP) of the major papillomavirus capsid protein L1 [30]. Since VLPs are merely protein and do not contain viral genome, these are considered non-infectious and non-oncogenic, and thus are safer than HPV-attenuated vaccines [31]. VLPs can be produced in bacteria, yeast, or insect cells. Cervarix comprises HPV16 and 18 VLPs, monophosphoryl lipid A (MPL), and aluminum hydroxide (together called adjuvant system 04, AS04) as an adjuvant [32]. MPL is a toll/like receptor 4 (TLR4) agonist that can induce high levels of antibodies as compared to Gardasil and Gardasil 9, both of which contain only aluminum hydroxide as an adjuvant and are produced in *Saccharomyces cerevisiae* yeast. Gardasil contains VLPs against HPV6, 11, 16, and 18, while Gardasil 9 contains VLPs against HPV6, 11, 16, 18, 31, 33, 45, 52, and 58 [33]. Table 1 shows the detailed comparison of the three vaccines available in the market.

The HPV vaccines currently being produced are based on L1-VLPs, which only provide type-restricted immunity, neglecting many other oncogenic HPV genotypes. Consequently, the second-generation VLPs, such as L2-VLP and Chimeric L1-L2 VLP, are drawing a lot of attention for their broader genotype coverage [34,35]. In comparison to L1-VLP, the minor capsid protein, L2, contains type-common epitopes that can provide broad cross-neutralizing antibody responses. Notably, Cervarix can confer a degree of cross-protection against some phylogenetically related types of HPV16 and 18 from the same phylogenetic cluster alpha-9 (HPV16-like: HPV31, 33, 35, 52, 58) and alpha-7 (HPV18-like: HPV39, 45, 59, 68) species groups, owing to its unique adjuvant systems [36].

## 4. HPV Vaccine Efficacy

Cervarix induces high anti-HPV16 and 18 antibody titers and can prevent the incidence of infection for at least 10 years [36,37]. In addition, Cervarix invokes a significantly high and long-term cross-reactive immunogenicity against HPV31 and 45. During a 10-year-follow-up study, over 85% of participants remained seropositive for anti-HPV31 and 45 antibodies following three doses of Cervarix [37]. In addition, Cervarix efficiently (>90%, injection prior to HPV exposure) protects against vaccine-targeted HPV related abnormalities and precancerous lesions, including cervical intraepithelial neoplasia 2 (CIN2), CIN3 and adenocarcinoma in situ (AIS) [38,39]. Cervarix also shows efficacy (>60%) in preventing all cervical precancerous lesions regardless of HPV infection or precancerous lesions caused by any HPV types [40,41]. Notably, Cervarix also showed strong protection against oral HPV16 and 18 infections. After a four-year vaccination period, a 93% reduction in the prevalence of oral HPV16 and 18 infections was reported [10]. 

Quadrivalent Gardasil shows excellent efficacy against cervical HPV infection, cervical cancer precursor lesions, and genital warts caused by the HPV types covered by Gardasil [9,11]. In addition, studies demonstrated that Gardasil significantly decreases HPV infections in the anus, vulva, and penis, as well as in the oral cavity related to HPV vaccine types [12,13,14]. Gardasil has a strong prevention rate (>90%, injection prior to HPV exposure) against CIN 2 or worse (CIN 2+), CIN 3+ and vulvar/vaginal intraepithelial neoplasia grade 2 or worse (VIN/VaIN 2+), caused by HPV 16 and 18 [6,41]. However, the inhibition on CIN 2 + and CIN 3 + caused by any HPV types was lower (20-50%) [6,41]. Comparatively, Gardasil demonstrated less cross-protection effect than Cervarix and the protection efficacy for HPV31, 33, 45, 52, and 58 were 46%, 29%, 7%, 18%, and 6%, respectively [36,42]. 

Gardasil 9 can efficiently prevent infections and cervical cancer precursor lesions (>95%, injection prior to HPV exposure) of any grade related to HPV types covered in the vaccine [15,43,44]. Gardasil 9 also showed around 90% and 80–85% inhibition in the incidence of vulvar and vaginal diseases, respectively [43,45,46]. A recent study reported that antibodies induced by Gardasil 9 could transfer across the placenta, which potentially protects the infant from HPV6 and 11 infections [47]. For HPV types not covered by the vaccine, Gardasil 9 has low cross-protective efficacy and has little effect on infections and diseases related to HPV types besides the nine vaccine types [44,48].

A recent systematic meta-analysis including 60 million individuals from 14 high-income countries showed that HPV vaccines significantly reduced the prevalence of HPV-related endpoints (genital HPV infections, anogenital wart diagnoses, or histologically confirmed CIN2+) among girls, women, and boys [49]. The most common HPV type, HPV16 and 18, significantly decreased by 83%, and HPV31, 33, and 45 decreased by 54%, among girls aged 13–19 years. The prevalence of anogenital warts decreased by 67%, and CIN2+ decreased by 51%, among girls aged 15–19 years. In addition to the significant decrease of HPV-related endpoints, herd effects among boys and older women were also observed in this meta-analysis [49].

## 5. Effects That Influence the Vaccine Coverage and Efficacy

There are many potential effects that influence vaccine coverage and efficacy, such as vaccine age, geographical regions, and education. The ideal time for the best protection against HPV-related diseases is prior to HPV exposure [50]. Studies demonstrated that vaccination before first sexual contact could protect more than 90% of targeted HPV-related infections, abnormalities, and precancerous lesions, while vaccination after HPV exposure only protects about 50–60% infections [38,40]. Taira et al. developed a disease transmission model and suggested that vaccination of 12-year-old girls provides the best and most cost-effective solution against cervical cancer [51]. The earlier the HPV vaccination is provided to the population before the sexual-behavior transition, the more effective the results are likely to be [52]. 

Studies showed HPV types in cancer and vaccine efficacy vary between geographical regions [45,53,54]. In squamous cell carcinoma, HPV16 was the predominant type, followed by HPV18, 45, 31, and 33 in many regions except Asia, where HPV58 and 52 were more frequently identified after HPV16 and 18 [54]. As many as 82% of invasive cervical cancers were related to HPV16 and 18 in Western and Central Asia, compared to 68% in Eastern Asia [53]. Studies found that Gardasil 9 protects against HPV-types-associated cervical cancers with an efficiency of 92% in Africa and North America, 91% in Europe, 90% in Latin America and the Caribbean, 88% in Asia, and 87% in Australia [45,55]. Moreover, ethnic disparities in HPV vaccinations were also found among United States- and foreign-born Hispanics and African Americans [56,57]. A meta-analysis suggested that ethnic minorities in the United States are more likely to initiate but less likely to follow through the full series of HPV vaccination [58]. 

Women’s knowledge and educational interventions prevent HPV transmission, increase the acceptability of HPV vaccine, and thus help to improve HPV vaccine coverage [59,60,61,62]. Video education has been reported to increase the willingness to accept the HPV vaccine by about 20% in young women [61]. Educational videos, outlining the risks of HPV and the benefits of vaccines have been shown to be an efficient way to improve vaccination behaviors [59,60]. In addition, a randomized trial with 30 pediatric and family medicine clinics showed that presumptive announcements led to a 5% increase in HPV vaccine coverage compared to control clinics [63]. A health care professional communication training intervention significantly improved HPV-vaccine-series initiation (10%) and completion (4%) [64]. All these results suggested a strong influence of educational interventions on HPV vaccine coverage. Therefore, further efforts should be put on promoting awareness of HPV as well as the protection and safety characteristics of HPV vaccinations. 

The number of sexual partners, one of the most important risk factors for HPV infection, has also been demonstrated as a strong predictor of CIN 2 and 3 regression [65,66,67,68,69]. Women with no lifetime or past-year sexual partners had significantly lower HPV vaccine initiation as compared with those with male sexual partners [70]. Notably, HPV vaccination did not show any comparative increase in sexual activity between vaccinated and unvaccinated men and women [71]. 

## 6. Male Vaccination

WHO guidelines recommend HPV vaccination focusing primarily on young girls, as females have 10 times higher risk of HPV-related cancers than males, and heterosexual males will be protected owing to herd immunity caused by high female vaccine coverage [19,72,73]. A study from the Netherlands showed that the risk of HPV-associated cancers among men could be reduced by 37% and 66% if vaccine uptake among girls reached 60% and 90%, respectively [72]. However, strong arguments emerged for extending vaccination to adolescent boys over the last decade [74]. Female HPV vaccine coverage in many countries is less than 60%, making the protection against males hard to confer. Moreover, indirect protection has minimal effect on homosexual men, putting them at a substantially higher risk of HPV infections and diseases [75].

Men have a higher risk of oral HPV infection and certain HPV-related cancers, which is associated with the number of lifetime oral sexual partners and tobacco use [76,77,78]. Around 20–30% of HNSCC and 50% or more tonsillar carcinoma contained HPV DNA [79,80,81,82,83,84]. Oncogenic oral HPV DNA was detected in 4% adults aged 20–69 years, while 8% of older men aged 50–59 years had oncogenic oral HPV infections [78]. The number of HPV-infection-induced cancer cases of oropharynx, oral cavity, and larynx in men were, respectively, 4, 2, and 7 times higher than female patients, and most of the cases were related to HPV types covered by the current vaccines [2,25]. Nevertheless, HPV infection is found in over 90% and 75% anal carcinoma cases among women and men, respectively [85,86]. The most common HPV types found in anal carcinoma are HPV16, 18, 31, 33, and 45, which are all covered by the current Gardasil 9 vaccine [85,86]. Some studies have suggested that HPV vaccination could protect against the progression of oral cancers, as oral HPV infection can be effectively inhibited by HPV vaccines [10,12,13]. However, this issue has not been fully addressed owing to the lack of alignment between vaccine availability and the changes in cancer development over time [87,88]. 

Pan-gender vaccination programs that extend to adolescent boys were implemented in a number of countries, including Australia, Austria, Bermuda, Brazil, Canada, Croatia, England, Germany, Israel, Italy, Lichtenstein, New Zealand, Norway, Serbia, Sweden, and the United States [89]. Some countries, such as England, provide free HPV vaccination to homosexual men too [90]. A study using the population-based single-type HPV transmission model with data from Sweden showed that the catch-up vaccination of males could lead to a reduction of about 17% in the HPV prevalence when compared with female-only vaccination [75]. Multiple models suggested that vaccinating both men and women is more beneficial in reducing HPV infections and diseases than vaccinating only females, although male vaccination has lower cost-effectiveness than female vaccination [72,75,91,92,93]. In countries where pan-gender vaccination programs are introduced, studies should be carried out to evaluate the impact of HPV vaccination on HPV-related cancers other than cervical cancer, especially among men.

## 7. Vaccine Safety and Adverse Effects

Multiple studies have demonstrated that all three vaccines exhibit excellent safety and tolerance in different age groups [94,95,96,97]. A 10-year-follow-up study showed that Gardasil is immunogenic, clinically effective, and generally well-tolerated in preadolescents and adolescents [98]. Furthermore, Cervarix and Gardasil 9 demonstrate great tolerance and antibody sustenance after vaccination for up to 9.4 years and 6 years, respectively [99]. The most frequent adverse effects (AEs) of Cervarix and Gardasil were injection-site reactions, such as pain and swelling, possibly due to the VLP-related inflammation process [100]. Cervarix can also lead to systemic symptoms, such as fever, nausea, vomiting, dizziness, myalgia, and diarrhea [100]. Headache and fatigue are the most common Cervarix-related systemic AE, seen in approximately 50–60% participants [101]. Gardasil and Gardasil 9 recipients may also have general symptoms, but no increased risk of systemic symptoms was evident in their recipients [100]. Some scientists proposed a hypothesis between HPV vaccination and small fiber neuropathy and dysautonomia [102]. HPV vaccination period was also found to overlap with post-vaccination symptoms, including chronic regional pain syndrome and autonomic and cognitive dysfunctions [103]. However, the same group later found that the vast majority of the girls that complained about unusual symptoms were initially diagnosed with psychiatric illness. Thus, no causal link has been demonstrated between HPV vaccination and the development of these symptoms [104]. Furthermore, several large meta-analysis studies found no significant observations of serious AEs, pregnancies, medically significant conditions, and new onset of autoimmune diseases after HPV vaccination [105,106]. A cohort study, including almost 1 million girls, following Gardasil vaccination showed that no serious AEs, such as autoimmunity, neurological, and venous thromboembolic AEs were identified when compared with background rates [107]. Although there were no serious AEs linked to HPV vaccination, more efforts are still needed to adjust the components, such as adjuvants, to mitigate the AEs, without hampering the vaccine efficiency.

## 8. Clinical Trials

National HPV vaccination programs were introduced in some countries more than 10 years ago, allowing surveillance trials to evaluate the long-term impact of vaccine introduction on community settings. A large number of clinical trials are focused on the effectiveness of HPV vaccines against infection and related cancers as well as optimization of the vaccine schedules [15,18,108,109]. In phase I-III clinical trials before marketing, all three vaccines showed excellent protection against vaccine-targeted HPV-related infection and pre-lesions of cervical cancer [44,101,110]. Gardasil and Gardasil 9 were efficacious against vaginal, vulvar, and anal dysplasia [11,44,111,112,113]. A large phase III trial study on Cervarix, called PATRICIA, demonstrated 100% protection against HPV 16 and 92.3% protection against HPV18 [114]. Furthermore, other studies on Cervarix have shown a high cross-protection against HPV31 and HPV45, with around 78% and 81% efficacy for protection against persistent HPV31 and HPV45 infection, respectively, and a 100% vaccine efficacy against CIN2+ or adenocarcinoma in situ caused by these two HPV types [114,115]. Gardasil clinical trials, called FUTURE I and FUTURE II, prevented 98% of HPV 16 and HPV18 related high-grade cervical lesions, if participants had not been previously exposed to either HPV 16 or HPV 18. Especially for HPV18, this vaccine demonstrated a 100% vaccine efficacy against CIN2+ [110,111]. The Gardasil 9 vaccine study (NCT00543543) showed 96% efficacy in preventing persistent infection and high-grade cervical, vulvar, and vaginal diseases related to HPV31, 33, 45, 52, and 58. Furthermore, it displayed equal efficacy as Gardasil to prevent diseases caused by HPV types 6, 11, 16, and 18 [44]. HPV vaccines also showed strong efficacy against oral HPV infections in a few studies, but whether HPV vaccination can inhibit the progress of HPV related oropharyngeal cancers has not yet been evaluated [10,12,13]. More clinical trials are needed to investigate the effect of HPV vaccine programs in inhibiting HPV associated oral, penile, anal, and vulvar diseases.

Herd protection was probably achieved in some countries and a decrease in the HPV prevalence among both vaccinated and non-vaccinated women was observed [27,109,116]. Post-licensure reports from countries with established national HPV vaccine programs demonstrated beneficial effects, at the population level, in the incidence of genital warts and high-grade cervical abnormalities [117,118,119,120]. Moreover, as the introduction of HPV vaccination in low- and middle-income countries remains the main challenge in tackling the burden of cervical cancer, alternative approaches such as one- or two-dose schedules are suggested to reduce economic challenges [121,122]. The comparative immunogenicity of two-dose and three-dose HPV-vaccine schedules are reported through antibody responses in young females [123,124]. 

The results of HPV infection during pregnancy were disputable. Some studies reported a risk of HPV infection, since the hormonal levels may change during pregnancy and alter the immune system [125,126]. Other studies showed that pregnancy does not influence HPV prevalence, incidence, and clearance [127,128,129]. A recent pooled analysis of clinical trials reported no differences in the prevalence of either oncogenic HPV types or HPV16 and 18 infections in women, before and after pregnancy [108]. More clinical trials are needed to evaluate the interactions between HPV infection and pregnancy.

## 9. HPV Vaccines for Therapy

Current HPV vaccines are prophylactic vaccines that do not treat pre-existing HPV infections and related conditions [44]. Researchers are working on therapeutic vaccines that trigger a cellular immune response as a treatment for established infections and malignancies [130]. There are large varieties of therapeutic vaccines, including bacterial-vector, viral-vector, peptide, nucleic acid, and cell-based vaccines, as well as combination therapies [131,132,133,134,135,136]. To date, unfortunately, with all the therapeutic vaccines tested, none of them could provide irreversible regression of HPV-associated cancers [137]. A randomized placebo-controlled clinical trial was approved recently to investigate the safety and efficacy of using a newly developed HPV type 16 E7-expressing *Lactobacillus*-based vaccine for the treatment of HPV16 positive HSIL [138]. Clinical trials that combined different therapeutic strategies, such as vaccines combined with a checkpoint inhibitor, showed promising results for the treatment of HPV-related cancers [139]. For example, combination therapy with antibodies against programmed death-ligand 1 (PD-L1) and HPV therapeutic vaccines have been reported to suppress tumor growth and increased immune-cell responses [140,141].

## 10. Future Prospects

HPV vaccines significantly decreased HPV infection and HPV related diseases. With the improvement in vaccine coverage and the introduction of pan-gender vaccination programs, better protection against HPV infections and fewer HPV-related cancer cases are expected. To achieve this, educational interventions introducing the risk of HPV and the benefits of vaccines are essential, especially in low- and middle- income counties. Decreased side effects with alternative adjuvants or other designs of vaccines will assist the acceptance and provision of vaccines at a young age. Another crucial problem is that the HPV types not covered by the vaccines are still at a high prevalence among young females [27,142]. The next-generation HPV vaccines should focus on high-valent vaccines with broad-protection spectrum. In addition, studies or clinical trials are essential to evaluate the impact of HPV vaccination on all HPV-related cancers. Therapeutic vaccines for cancer treatments are of great importance and entail a promising future aimed at combating HPV infection and related diseases from prevention to clearance.

## Figures and Tables

**Table 1 vaccines-08-00391-t001:** Comparison of HPV vaccines.

HPV Vaccines	Cervarix		Gardasil	Gardasil 9
Time of FDA Approval	2009		2006	2014
Manufacture	GSK		Merck & Co	Merck & Co
VLP Types	6	-	20 µg	30 µg
	11	-	40 µg	40 µg
	16	20 µg	40 µg	60 µg
	18	20 µg	20 µg	40 µg
	31	-	-	20 µg
	33	-	-	20 µg
	45	-	-	20 µg
	52	-	-	20 µg
	58	-	-	20µg
Expression system	Baculovirus -Insect Cell		Yeast	Yeast
Adjuvant	50 µg MPL absorbed on 500 µg aluminum hydroxide (AS04)		225 µg aluminum hydroxyphosphate sulfate	500 µg aluminum hydroxyphosphate sulfate
Dose	0.5 mL/dose		0.5 mL/dose	0.5 mL/dose
Injection schedule	0, 1, 6 months		0, 2, 6 months	0, 2, 6 months
Cervical cancer Protection rate	70%		70–75%	90%

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
