# Peer review of "Human Papillomavirus Vaccines: An Updated Review"

_vaccines, 2020, doi:10.3390/vaccines8030391_

Round 1

Reviewer 1 Report

Dear Authors,

the manuscript submitted treats a very interesting and current topic. However, some aspects should be addressed before to consider the paper suitable for potential publication. 

Please find below my recommendations:

  • The authors split the document in more section. This seem well but in section 7 the did not dicuss the role (or not) that some authors describe on the onset of some AEs such as autoimmune ones (especially neurological ones) given by HPV vaccines. Please add to the article.
  • Moreover, section 8 seems too short. Please add more references and describe the main results of clinical trial.

Author Response

The manuscript submitted treats a very interesting and current topic. However, some aspects should be addressed before to consider the paper suitable for potential publication. 

Please find below my recommendations:

  • The authors split the document in more section. This seem well but in section 7 the did not dicuss the role (or not) that some authors describe on the onset of some AEs such as autoimmune ones (especially neurological ones) given by HPV vaccines. Please add to the article.

Response: Thanks for pointing out this important information. Now we have added references about AEs includingautoimmune. See updated new manuscript line 214 to line 225.

  • Moreover, section 8 seems too short. Please add more references and describe the main results of clinical trial.

Response: Thanks for this comment. Now we have added references and result in the clinical trial section. See updated new manuscript line 232 to line 257.

Reviewer 2 Report

This is a well-organized, well-written, and thoroughly referenced review that covers the major HPV vaccine issues. I have no suggestions other than to correct some minor sentence errors listed below.

  1. Line 62: insert the word "of" between "80%" and "cervical".
  2. Liine 80: the number "2" is missing from the phrase "Chimeric L1-L VLP".
  3. Line 101: insert the word "the" before anus.
  4. Line 102: insert the word "the" before oral.
  5. Line 173: insert the word "of" between "8%" and "older".
  6. Line 178: the word "comment" should be "common".
  7. Line 226: the "neither/nor" in this sentence should be "either/or".
  8. Line 236: the word "Till" should be replaced with "To".
  9. Line 238: the word "is" should be replaced with "was".
  10. Line 246: insert the word "and" between the words "infection" and "HPV".

Author Response

  • This is a well-organized, well-written, and thoroughly referenced review that covers the major HPV vaccine issues. I have no suggestions other than to correct some minor sentence errors listed below.
  1. Line 62: insert the word "of" between "80%" and "cervical".
  2. Liine 80: the number "2" is missing from the phrase "Chimeric L1-L VLP".
  3. Line 101: insert the word "the" before anus.
  4. Line 102: insert the word "the" before oral.
  5. Line 173: insert the word "of" between "8%" and "older".
  6. Line 178: the word "comment" should be "common".
  7. Line 226: the "neither/nor" in this sentence should be "either/or".
  8. Line 236: the word "Till" should be replaced with "To".
  9. Line 238: the word "is" should be replaced with "was".
  10. Line 246: insert the word "and" between the words "infection" and "HPV".

Response: Thanks for pointing out the gramma mistakes. Now we have updated all the listed words as suggested.